# Continuous Particle Separation Driven by 3D Ag-PDMS Electrodes with Dielectric Electrophoretic Force Coupled with Inertia Force

**DOI:** 10.3390/mi13010117

**Published:** 2022-01-12

**Authors:** Xiaohong Li, Junping Duan, Zeng Qu, Jiayun Wang, Miaomiao Ji, Binzhen Zhang

**Affiliations:** 1Key Laboratory of Instrumentation Science & Dynamic Measurement Ministry of Education, Micro Nano Technology Research Center, North University of China, Taiyuan 030051, China; xiaohongli198608@163.com (X.L.); duanjunping@nuc.edu.cn (J.D.); zqu@nuc.edu.cn (Z.Q.); wangjiayun@nuc.edu.cn (J.W.); S2006240@st.nuc.edu.cn (M.J.); 2Taiyuan Institute of Technology, Taiyuan 030051, China

**Keywords:** dielectrophoresis, 3D electrodes, inertial, microfluidic chip, particle sorting

## Abstract

Cell separation has become @important in biological and medical applications. Dielectrophoresis (DEP) is widely used due to the advantages it offers, such as the lack of a requirement for biological markers and the fact that it involves no damage to cells or particles. This study aimed to report a novel approach combining 3D sidewall electrodes and contraction/expansion (CEA) structures to separate three kinds of particles with different sizes or dielectric properties continuously. The separation was achieved through the interaction between electrophoretic forces and inertia forces. The CEA channel was capable of sorting particles with different sizes due to inertial forces, and also enhanced the nonuniformity of the electric field. The 3D electrodes generated a non-uniform electric field at the same height as the channels, which increased the action range of the DEP force. Finite element simulations using the commercial software, COMSOL Multiphysics 5.4, were performed to determine the flow field distributions, electric field distributions, and particle trajectories. The separation experiments were assessed by separating 4 µm polystyrene (PS) particles from 20 µm PS particles at different flow rates by experiencing positive and negative DEP. Subsequently, the sorting performances of the 4 µm PS particles, 20 µm PS particles, and 4 µm silica particles with different solution conductivities were observed. Both the numerical simulations and the practical particle separation displayed high separating efficiency (separation of 4 µm PS particles, 94.2%; separation of 20 µm PS particles, 92.1%; separation of 4 µm Silica particles, 95.3%). The proposed approach is expected to open a new approach to cell sorting and separating.

## 1. Introduction

Since the proposal of the concept of microfluidic chips by Manz and Widmer et al. [1], microfluidic techniques using fluid as a medium have received increasing attention. They offer the potential advantages of reduced sample consumption [2,3], high sensitivity, and ease of mass production compared with traditional separation techniques, such as centrifugation and filtration. Microfluidic technology is now widely used in medical diagnosis [4], biological detection [5], chemical analysis [6], and other aspects where particle and cell separations are critical to numerous applications. Many techniques have been developed in microfluidics, including inertial microfluidics [7], deterministic lateral displacement [8], hydrophoresis [9,10], optical [11], dielectrophoresis (DEP) [12], surface acoustic waves [13], and magnetic force to achieve precise control and sorting of detection objects such as particles and cells with microfluidic chips. Among these separation techniques, DEP has attracted more attention due to its advantages, such as label-free and non-contact forces on particles [14,15]. In DEP, the internal charge of particles in fluid is induced to polarize and move in the positive or negative direction of the electric field gradient after the particles are loaded with the non-uniform electric field [16]. Currently, DEP sorting generally uses a sheath flow focused on using planar electrodes. These electrodes are generally in the form of a thin-film metal layer at the bottom of the microchannel, and the electric field intensity is exponentially attenuated as the vertical distance from the electrode increases. Moreover, as the particle moves further away from the planar electrode, the electrophoretic force on the particle decreases rapidly. Therefore, the DEP force features a limited range of particle manipulation, and particles are easily absorbed in the edge of the electrode or channel surface owing to the strong partial electric field, resulting in particle damage.

Particles dispersed at different heights can be affected by the electric field force to expand the action area of the DEP effect. Hence, the application of 3D electrodes in microfluidic devices is an effective method. Three-dimensional electrodes feature the same height as the microfluidic chip channel. This can provide a non-uniform electric field in the vertical direction, thus improving the attenuation of the two-dimensional electric field in the vertical direction compared with the DEP device with a thin planar electrode, as well as producing a greater sorting efficiency and increasing the throughput. Jie Yao et al. fabricated 3D carbon electrodes via screen-printing to complete the sorting of blood cells from circulating tumor cells [17]. Jia et al. designed 3D Ag-polydimethylsiloxane (PDMS) electrodes filling into the mold, which was laminated onto the glass substrate by two layers of a negative dry film to obtain a 60 µm photoresist layer, and eventually separated Au-plated polystyrene particles and yeast cells [18]. Fu et al. fabricated 3D electrodes composed of nanosized carbon black and PDMS on both sides of the separation channel to generate regional electrophoresis and isokinetic electrophoresis [19]. Among these 3D electrode preparation methods, the Ag-PDMS composite conductive material has been widely used due to its advantages of low cost, simple production, and better conductivity. However, most different cells feature similar dielectric properties, resulting in poor sorting performance. Therefore, active separation coupled with passive separation has become a simple and efficient method of particle separation [20]. Inertial separation, as the passive sorting method, uses inertial force within the fluid to deflect the trajectory of cells and produce continuous and high-throughput cell separation without applying external forces [21,22]. Commonly used inertial sorting channels include spiral channels [23,24], curved channels [25,26], and contraction-/expansion(CEA) structures [27,28]. The CEA structure can achieve sorting according to the size of the cells or particles at a low Reynolds number, preventing damage to the cells or particles from high shear forces in conventional inertial sorting.

In this study, a microfluidic chip combining the effects of 3D Ag-PDMS electrodes [29] on both sides of the main channel and microfluidic channel with a CEA structure was designed and fabricated to produce continuous particle separation. The particle focusing and sorting were achieved by focusing the particles with a combination of DEP force and inertial force. The designed structure of the microfluidic chip was first simulated using commercial software COMSOL5.4. The simulations and analysis were mainly focused on the flow field distribution, the electric field distribution, and the motion state of the particle trajectory. Next, a composite conductive material, Ag-PDMS, was selected for the 3D electrode tests to verify the sorting performance of the structure. In the experiment, a spatially non-uniform electric field was generated by energizing the 3D electrode with an alternating current (AC) sinusoidal voltage to extend the DEP effect to different heights of the main channel. The separating experiments on three kinds of particles, 4 µm polystyrene, 20 µm polystyrene, and 4- µm silica, were conducted. The recovery of each particle was 94.2%, 92.1%, and 95.3%, respectively.

## 2. Materials and Methods

### 2.1. Theory (DEP and Inertial Force)

The DEP force is a phenomenon in which particles suspended in solution are polarized in a non-uniform electric field and move with the fluid [30]. The direction and magnitude of the DEP force depend on the difference in polarization between the particles and the surrounding medium. It can be described as follows [31]:(1)$$F_{DEP}=2\pi r^{3}\varepsilon_{m}R_{e}\left[K\left(\omega \right)\right]\nabla E_{rms}^{2}$$
where *r* is the radius of the particles, $\varepsilon_{m}$ is the permittivity of the suspension liquid, $\nabla E_{rms}^{2}$ is the gradient of the square of the applied electric field, and $R_{e}\left[K\left(\omega \right)\right]$ refers to Clausius–Mossotti (CM) factor [32], where [K(ω)] can be denoted as [33]:(2)$$K\left(w\right)=\frac{\varepsilon_{p}^{*}-\varepsilon_{m}^{*}}{\varepsilon_{p}^{*}+2\varepsilon_{m}^{*}}$$
where *p* denotes particles, m denotes medium, $\varepsilon^{*}=\varepsilon -\frac{i\sigma }{w}$ is the complex permittivity, σ is the electrical conductivity, $i=\sqrt{-1}$, and ω is the angular frequency of the electric field, w=2πf. For spherical particles, the conductivity can be expressed as the sum of the surface conductivity of the bulk and surface conductivities according to the study by O’Konski [34].
(3)$$\sigma_{p}=\sigma_{b}+\frac{2K_{s}}{r}$$
where $\sigma_{b}≅0,\;Ks$ is a general surface conductance (typically 1nS for latex particles), and r  is the radius of particles.

When the polarization of particles is above the polarization of the dielectric solution, the particles are subjected to the positive dielectric force pDEP and move towards the high electric field region; by contrast, it is subjected to negative dielectric force (nDEP). In addition, the particles in the microfluidic channel are subjected to fluid drag force due to the flow of fluid, which can be expressed as: $F_{drag}=6\pi \eta rv$, where η is the viscosity of the fluid, v is the velocity of the fluid.

Inertial migration occurred when particles were dispersed in tubular flow with finite inertia [35]. In Newtonian fluids, the fluid near the wall is subject to frictional forces due to the laminar flow of the fluid, which impedes the movement of the fluid and thus leads to a parabolic distribution of the flow velocity in the channel [36]. This phenomenon creates a shear gradient, which induces a shear-induced lift force that pushes particles suspended in the fluid toward the channel wall [37]. The wall-induced lift force pushes the particles to the center of the flow channel as the particles move close enough to the channel wall. Finally, the combined force in opposite directions is called inertial force, which can be expressed as follows [38]:(4)$$F_{L}=\frac{\rho_{f}r_{p}^{4}v_{m}}{D_{h}}C_{L}$$
where $\rho_{f}$ is the fluid density, $v_{m}$ is the flow velocity, $r_{p}$ is the particle diameter, $D_{h}$ is the hydraulic diameter of the expansion region, and $C_{L}$ is the lift coefficient, which depends on the Reynolds number and the position of the particle on the channel cross-section. Particles achieve inertial focusing where the inertial lift force $F_{L}=0$ in the channel cross-section.

### 2.2. Design of the Microfluidic Separation Chip

The proposed microfluidic chip consisted of an upper PDMS microchannel, a microchannel sidewall Ag-PDMS and a lower Indium Tin Oxides (ITO) transition electrode. The PDMS microchannel was adopted as the contraction and expansion channel [35]. The Ag-PDMS electrodes covered both ends of the PDMS microchannel sidewalls with the same height as the microchannel to ensure that the non-uniform electric field covered the whole channel. In addition, the length of the electrodes and contraction/expansion microchannels were equal in this microchip to ensure the same processing time of DEP separation and inertial separation. The PDMS microchannel part contained the main channel and four branches that led to one inlet and three outlets. All the microchannels in the devices were 60 µm in height. The structure of the designed microfluidic chip is shown in Figure 1A. The schematic diagram of the microchannel and 3D electrode structure is shown in Figure 1B.

After passing through the inlet, different particles flowed through the trapezoidal contraction channel, which caused a sudden change in velocity so that the particles focused on the top of the channel. In addition, after a sinusoidal voltage of equal magnitude and opposite direction was applied on the two side walls of the channel, the nonuniform electric field consistent with the height of the channel was generated in the channel. Therefore, the electric field did not decay with the increase in the height of the channel, thus ensuring that the nonuniform electric field covered the whole channel. By adjusting the voltage frequency and the conductivity of the solution, the 20 µm polystyrene (PS) particles experienced pDEP, the 4 µm PS particles experienced weaker pDEP, and the 4 µm silica particles experienced nDEP. Finally, this device could perform the high-efficiency sorting of multiple particles through the combination of dielectrophoretic force, inertial force, and fluid traction during the practical experiments.

### 2.3. Fabrication of the Microfluidic Chip

The microfluidic chip fabrication process was mainly divided into two parts: one for the preparation of microchannels and 3D electrodes and another for the preparation of ITO electrodes. The fabrication of the PDMS channel was mainly based on soft lithography [39]. Next, 3D electrodes were prepared. First, PDMS and curing agents in a ratio of 10:1 were fully mixed and their bubbles were removed by the vacuum chamber to form a PDMS curing agent. Next, the high-purity micron silver powder was mixed with the PDMS curing agent in a mass ratio of 86:14. After being fully mixed, the Ag-PDMS was placed in the vacuum chamber to remove bubbles for 1 h. Subsequently, it was applied to the silicon wafer and smoothened with a spatula. The PDMS (prepolymer: curing agent = 10:1) was poured on this mold at 75 °C for 50 min and then peeled off gently after cooling with Ag-PDMS. The Ag-PDMS was heated to 150 °C to ensure good electrical conductivity. A transition electrode was needed to connect the 3D electrode to the external wire, which was prepared by the wet etching method using ITO glass. The ITO electrode preparation process is similar to the silicon process. After post-baking, the ITO electrode preparation process was similar to the silicon process. After baking, the ITO glass was placed in the etchant with a solution ratio of 50:50:3 in the order of H_2_O: HCl: HNO_3_ at 55 °C for 100 s. Next, the ITO glass was placed in acetone and ultrasonicated for 5 min to remove the covered positive PR and then placed in ethanol for ultrasonication for 5 min. Subsequently, ITO glass was washed with a large amount of deionized water and dried with nitrogen gas.

Eventually, the surface of the PDMS channel with 3D electrodes and ITO glass was bonded by a plasma process. Figure 2 shows the fabrication process.

### 2.4. Sample Preparation and System Setup

The size of some biological cells was close to 5–20 µm. Polystyrene (PS) microspheres (BaseLine, 2.5 wt%) with diameters of 4 µm and 20 µm and 4 µm silica particles (BaseLine, 4 wt%) were chosen as experimental samples for this test to better simulate real cells. To prepare the PS solution, 2 mL of 4 µm PS suspension and 1 mL of Tween 20 were made up to 6 mL with water and ultrasonicated for 5 min. Next, the 20 µm PS microspheres and 4 µm silica microspheres were prepared by the same method. The two suspensions were mixed in a ratio of 1:1 to produce two sample solutions, which were a mixture of 4 µm silica microspheres and 20 µm polystyrene particles [40]. Further, a mixture of three particles, 4 µm silicon microspheres, 4 µm polystyrene particles, and 20 µm polystyrene particles, was produced in a ratio of 1:0.5:1. Subsequently, phosphate-buffered saline solution was added to adjust the conductivity and provide solutions with conductivities of 0.1 µS cm^−1^, 1 µS cm^−1^, 4 µS cm^−1^, and 10 µS cm^−1^. Each type of test was performed at least three times. The particle separation device was mainly composed of a computer, an inverted microscope (Olympus CKX53), a microinjection pump (Harvard, Holliston, MA, USA), and the prepared microfluidic chip. The velocity of the particles was controlled with a micro-syringe pump and the trajectories of the particles were visualized using the inverted microscope.

## 3. Results and Discussion

### 3.1. Numerical Simulation Results

In this study, numerical simulations were performed using the commercial finite element software, COMSOL5.4, to obtain the effects of flow velocity distribution, electric field distribution, and particle motion trajectory distribution. We first used COMSOL to set the inlet phase flow rate as 3 mm/s. The actual flow rate was calculated as 3 µL/min because the height of the prepared microfluidic chip was 60 µm.

Figure 3A, B illustrates the distributions of the flow field in different parts, and the arrows indicate the direction of the flow velocity. We found that the change in the flow velocity in the middle of the microchannel was the largest, especially at the center of the contraction channel. The flow rate at the center of the contraction region was about twice that of the expansion region. However, the flow rate on both sides of the channel was close to zero. Therefore, the fluid motion in the width direction of the microchannel was in a parabolic form, implying that it conformed to the laminar flow. Next, the a–a’ cross-section of the central part of the CEA channel was selected to analyze the variation in the flow velocity, as shown in Figure 3D. The velocity of the fluid increased abruptly when the fluid flowed from the expansion channel into the contraction channel. Since the inertial force and fluid drag force were affected by the particle diameter and fluid flow velocity, the contraction/expansion structure made the flow line bend strongly when particles of different sizes entered the trapezoidal channel contraction structure with the fluid. Eventually, different particle sizes could be focused in different positions. Figure 3C illustrates the distribution of the electric field. COMSOL 5.4 was used to set the boundary conditions so that one side of the channel featured high potential and the other side featured low potential. All the boundaries were selected except one inlet and three outlets due to the 3D electrodes. Figure 3E shows the electric field distribution of the b–b’ cross-section in the central part of the selected CEA channel. The electric field was higher in the contraction region than in the expansion region. When the particles moved through the contraction region, they were exposed to the area of a relatively high electric field; however, they were affected due to the fast flow rate, which made the particles last for a short time at the applied voltage of 5 V and frequency of 10 kHz.

The trajectory of the particles was affected except by the dielectrophoretic force and particle size. It also depended on the Clausius–Mossotti (CM) factor value, which varied with the conductivity and dielectric properties of the particles and dielectric solution, as well as the frequency of the applied electric field. The CM factor value was calculated and simulated using the Matrix Laboratory (MATLAB) program to obtain three kinds of particle at different frequencies and solution conductivities, as shown in Figure 4A–C. Figure 4A shows that when the solution conductivity was 4 µS/cm, the CM factor was zero and the 4 µm polystyrene spheres were not affected by the dielectrophoretic force; when the solution conductivity was higher than 4 µS/cm, no matter how the applied frequency changed, the dielectrophoretic force was negative. If the solution conductivity was smaller than 4 µS/cm, the particles within 100 kHz were subjected to positive dielectrophoretic force, and vice versa, by negative dielectrophoretic force. Figure 4B illustrates the variation with the direction and magnitude of the dielectrophoretic force on the 4 µm silica spheres when the conductivity of the solution or the frequency of the electric field changed. Due to the low conductivity of silica, no matter how the conductivity of the solution varied, the particles were subjected to negative DEP force in the nonuniform electric field. Figure 4C shows the CM factor of the 20 µm polystyrene spheres with the change in solution conductivity and frequency. It was observed that the particles within 20 kHz were subjected to positive dielectrophoretic forces only at a conductivity of 0.1 µs/cm.

Finally, we concluded that when the conductivity was 0.1 µs/cm, the PS beads with 20 µm and 4 µm particle sizes moved laterally in the direction of high electric field intensity when passing through a non-uniform electric field due to the positive value of CM factor. The 4 µm silica microspheres were influenced by negative dielectrophoretic force and moved in the direction of low electric field intensity.

The simulation of the cell trajectory is shown in Figure 5A–C. The ratio of the 20 µm PS beads, 4 µm PS beads, and 4 µm silica particles was set as 1:1:1. After applying electric signals with a 5 V voltage and 10 kHz frequency, the 20 µm and 4 µm PS particles flowed into two bilateral outlets of the microchannel due to the combination of three forces: pDEP, drag force, and inertial force. The 4 µm silica particles focused in the center of the channel and finally flowed out of the intermediate outlet. Nevertheless, the simulation results demonstrated clearly that this structure separate three types of particle.

### 3.2. Discussion

The separation of particles by size and dielectric properties was tested to verify the performance of the separation device.

First, a mechanical analysis was performed. As can be seen in Figure 6A, B, when no electric signal was applied, particles in the microchannel mainly experienced inertial fore $F_{L}$ and drag force $F_{D}$. According to Equation (4) and the formula of fluid drag force $F_{drag}=6\pi \eta rv$, it can be concluded that $F_{L}\propto a^{4},\;F_{D}\propto a$. Thus, as the particle size increased, the growth rate of the inertial force was much higher than that of the fluid drag force. However, at low velocities, both the inertial force and the fluid drag force on the particle featured the same order of magnitude on the particles, which resulted in neither force being dominant. The 4 µm and 20 µm particles moved randomly in the microchannel. After applying the electric signal, both particles were also subjected to DEP force $F_{DEP}$. Due to the repeated CEA microchannels, the electric field gradient increased; thus, the DEP force increased (according to Equation (1)). Owing to the combination of DEP force, inertial force, and drag force, the physical differences between the 4 µm particles and the 20 µm particles were amplified. Therefore, superior particle separation can be achieved according to the differences in dielectric properties and particle size at low velocities.

Next, the performance was tested under low flow rate conditions with particles of 4 µm and 20 µm PS spheres suspended in a mixture with a solution conductivity of 4 µS/cm, without and with voltage applied for particle separation. When the particle velocity increased, the time required for the particles to pass through the main channel decreased, and the DEP action time decreased. Consequently, when loading the 3D electrode, the particle flow rate should not be high. Therefore, the particle flow rates to test the particle sorting effect in both the case of no AC voltage and of loaded voltage were 3 and 14 µL/min. When no AC voltage was applied to the electrodes, the particles showed a random motion in the channel at a flow rate of 3 µL/min. When the flow rate increased to 14 µL/min, the sorting effect of the two particles was not obvious, although the particles were subjected to inertial force under the action of the CEA channel.

When a voltage of 5 V was loaded onto the electrodes, a three-dimensional, non-uniform electric field was generated in the main channel. At this point, the particles could be subjected to dielectrophoretic forces at different channel heights. The performance of the device was tested for particle sorting under pDEP and nDEP. When the particle velocity was 3 µL/min, voltages of 5 V and −5 V at 10 kHz were applied to the 3D electrodes on the two side walls. Figure 7A–E shows the experimentally observed images at different flow velocities. When the mixed particles in the medium entered the main channel through the inlet, the particles were focused after passing through the trapezoidal-shaped constricted and expanded channel under the combined effect of inertial and dielectrophoretic forces. The 4 µm particles experiencing pDEP in the suspended medium were deflected in the direction of the upper and lower electrodes and then flowed out from outlet 1 and outlet 3. Meanwhile, the 20 µm particles experiencing nDEP flowed out from outlet 2.

Figure 8A, B shows the separation efficiency with different solution conductivities and flow rates, respectively, at outlet l. With a conductivity of 1 µS cm^−1^, the separation efficiency was 98%. However, when the conductivity was dramatically increased to 4 µS cm^−1^, the separation efficiency decreased. The CM factor of the 4 µm PS particles was zero. The fraction of the 4 µm PS particles flowed into outlet 2, which led to a decrease in the number of 20 µm PS particles. When the solution conductivity was 10 µS cm^−1^, both the particles were subjected to the nDEP force. Hence, the 20 µm PS particles that flowed into outlet 2 were extremely low. The selected conductivity should be higher than 1 µS cm^−1^. When the flow rates changed from 3 µL min^−1^ to 50 µL min^−1^, the separation efficiency change was not very obvious. When the flow rate was 14 µL min^−1^, both the separation efficiency (98.2%) and the sorting throughput were very high. Figure 8C, D indicates the separation efficiency of outlet 1 and outlet 2. When the solution conductivity changed from 3 µL min^−1^ to 10 µL min^−1^, the separation efficiency also decreased due to the influence of the solution conductivity on the CM factor. When the flow rates were 3 µL min^−1^ and 14µL min^−1^, the separation efficiency and the sorting throughput were also very high.

Figure 9A, B shows the sorting efficiency with inertial force only and a combination of DEP force and inertial force when the flow rates changed from 3 µL min^−1^ to 50 µL min^−1^ at outlet 2. When no electric signal was applied, the sorting efficiency was relatively low.

At a solution conductivity of 0.1 µS/cm, three particles (4 µm and 20 µm PS spheres and 4 µm silicon dioxide microspheres) were suspended in the medium and moved randomly into the main channel. At this time, both 4 µm and 20 µm polystyrene spheres were subjected to pDEP; the radius of the 20 µm particles was four times that of the 4 µm particles. Thus, under the joint action of the DEP force, inertial force, and drag force, the 20 µm particles were subjected to a much larger force than the 4 µm particles. Therefore, the 20 µm particles were deflected towards exit 1, and the 4 µm particles were deflected to exit 3; the 4 µm silica particles were deflected to exit 2 due to the nDEP’s action. Appendix A can be found in the Appendix A. Figure 10A–C shows the separation process of the three kinds of particles. The sorting efficiency for the 4, 20, and 4 µm particles was more than 90%.

## 4. Conclusions

In this study, a new micro-device using 3D electrodes and contraction expansion channels achieved particle separation. The use of 3D electrodes increased the non-homogeneity of the electric field more compared with the traditional flat electrodes; hence, the particles were subjected to dielectrophoretic forces at different heights. Besides the inertial force generated by the contraction and expansion of the channel, particle sorting was mainly dependent on the regulation of the conductivity of the solution and the frequency of the electric field loaded on the 3D electrode to achieve efficient sorting. The particle flow velocity distribution, electric field distribution, and particle motion trajectory were obtained through a COMSOL software simulation. The particle sorting experiments were first conducted for 4 µm and 20 µm PS spheres. When the solution conductivity was 1 µS/cm, the 4 µm PS spheres were subjected to pDEP and the particles were deflected in the direction of the upper and lower electrodes and exited from outlet 1 and outlet 3. Meanwhile, the 20 µm particles that experienced nDEP exited from outlet 2. In addition, the mixture experiments of the three particles were successfully performed using different principles for the sorting of the three particles. The experimental results showed that the device could perform the high-precision sorting of particles. Based on the work and analysis, we believe that the proposed method can be used in medical detection and drug screening and promote the development of highly integrated chip systems. Furthermore, we also hope this device can be applied to CTC separation in clinical applications.

## Figures and Tables

**Figure 1 micromachines-13-00117-f001:**
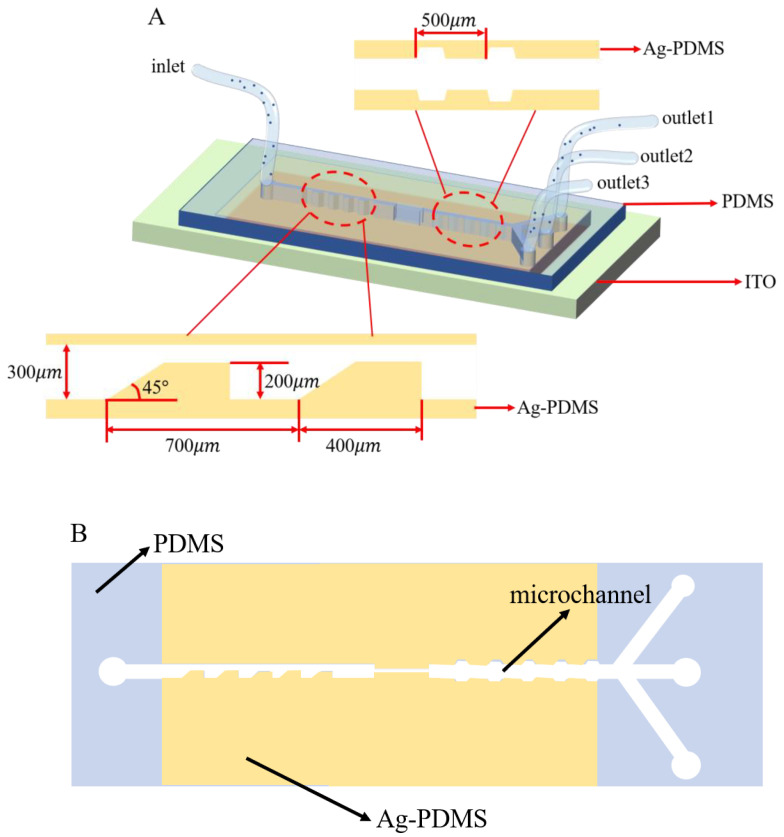
(**A**) Design of the microchip in the particle separation device. The micro-device consisted of one inlet, the CEA channel region, and three outlets. (**B**) Schematic diagram of Ag-PDMS electrode layer.

**Figure 2 micromachines-13-00117-f002:**
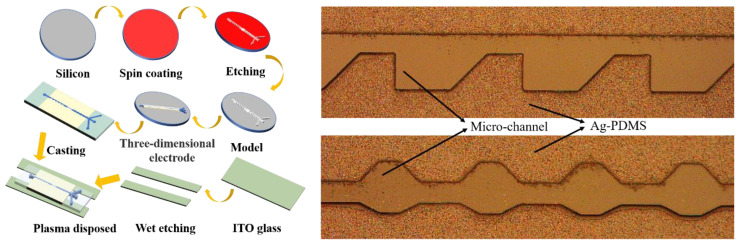
(**A**) Fabrication process of the microfluidic chip; (**B**) confocal microscopy image of the 3D electrodes.

**Figure 3 micromachines-13-00117-f003:**
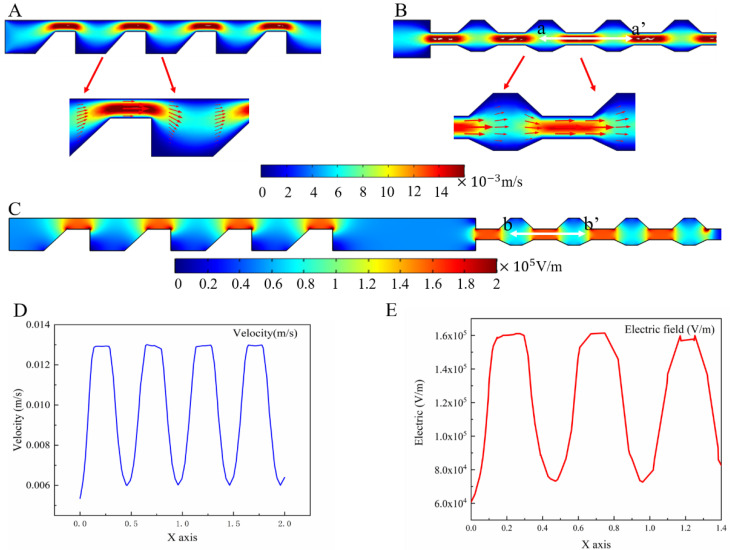
(**A**) Distributions of the flow field in contraction and expansion of the entrance part; (**B**) distributions of the flow field in contraction and expansion of the outlet part; (**C**) distribution of the electric field; (**D**) a–a’ cross-section of the central part of the CEA channel; (**E**) electric field distribution of b–b’ cross-section.

**Figure 4 micromachines-13-00117-f004:**
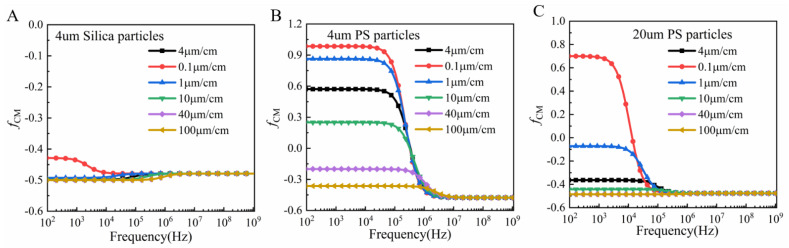
(**A**) CM factor of the 4 µm polystyrene spheres. (**B**) CM factor of the 4 µm silica particles. (**C**) CM factor of the 20 µm polystyrene spheres.

**Figure 5 micromachines-13-00117-f005:**
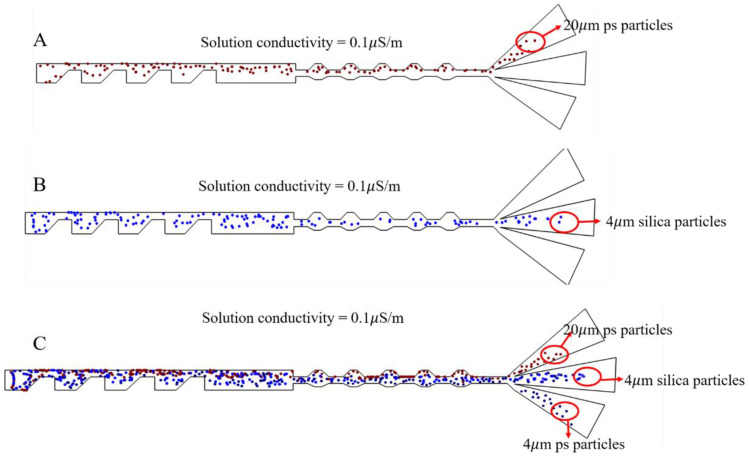
(**A**) Particle trajectory simulation of 20 µm polystyrene spheres. (**B**) Particle trajectory simulation of 4 µm silica particles. (**C**) Particle trajectory simulation of 20 µm polystyrene spheres, 4 µm silica particles, and 4 µm polystyrene spheres.

**Figure 6 micromachines-13-00117-f006:**
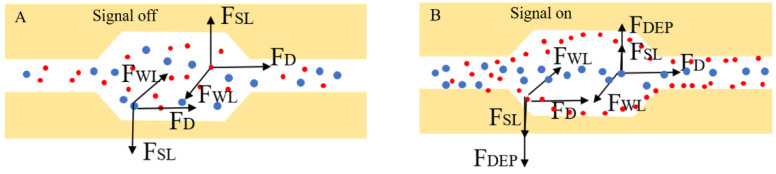
(**A**) The mechanical analysis when no signal was added (inertial sorting); (**B**) the mechanical analysis when the signal was applied (coupled with inertial force and DEP force).

**Figure 7 micromachines-13-00117-f007:**
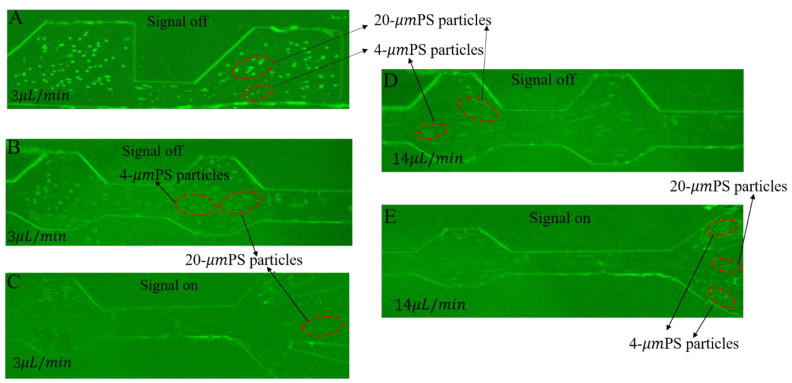
(**A**) Particle trajectory near the inlet at 3 µL/min without signal applied. (**B**) Particle trajectory near the outlet at 3 µL/min without signal applied. (**C**) Particle trajectory at 3 µL/min with 5 V signal applied. (**D**) Particle trajectory at 14 µL/min without signal applied; (**E**) Particle trajectory at 14 µL/min with 5 V signal applied.

**Figure 8 micromachines-13-00117-f008:**
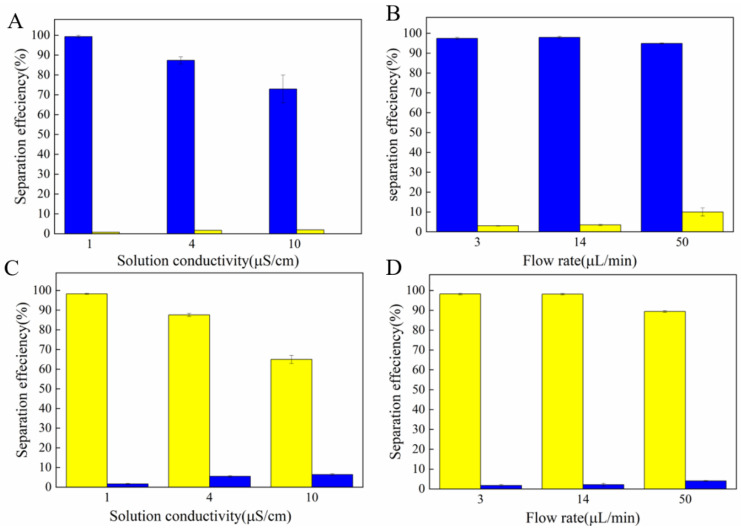
Statistics of separating efficiency under the parameters of different solution conductivities and flow rates. (**A**,**B**) Middle channel; (**C**,**D**) branch channels.

**Figure 9 micromachines-13-00117-f009:**
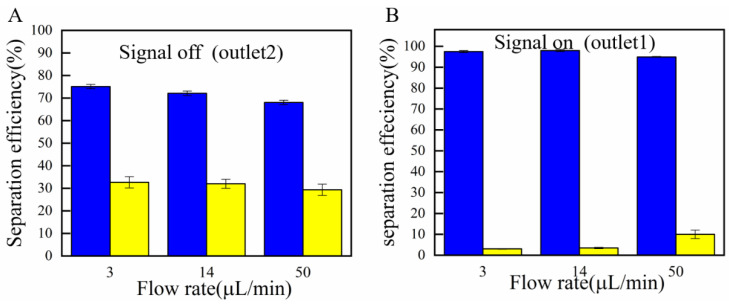
(**A**) The separation efficiency with pure inertial force. (**B**) The separation efficiency with combination of DEP force and inertial force.

**Figure 10 micromachines-13-00117-f010:**
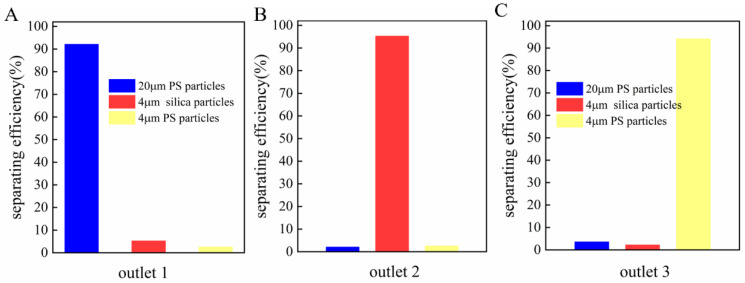
(**A**–**C**) Continuous separation of three kinds of particles in the middle channel.

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
