# Peer review of "Continuous Particle Separation Driven by 3D Ag-PDMS Electrodes with Dielectric Electrophoretic Force Coupled with Inertia Force"

_micromachines, 2022, doi:10.3390/mi13010117_

Round 1

Reviewer 1 Report

Continuous particle separation driven by 3D Ag-PDMS electrodes with dielectric electrophoretic force coupled with inertia force

The authors provide an interesting device in this manuscript. The manuscript is well written. However, I have some feedback about the experiments and the device. Here are my comments:

  1. The authors have shown their device in Figure 1, however, it is not clear how the design parameters for this device were obtained. Please also explain the significance if the structures used in this device and provide references for the designs.
  2. Fabrication method: Provide references for fabrication of Ag-PDMS channel and ITO electrodes.
  3. Sample preparation: Line 190-191, how did you select the particle concentration for this experiment. How does this affect the efficiency?
  4. The effect of inertial forces in your device is not clear. The 20 um particles show high pDEP, 4 um polystyrene particles show low pDEP and 4 um silica particles show nDEP, therefore I believe that the difference in DEP alone can be used to sort these particles. Please explain how inertial forces are contributing to your device. Providing scale and comparison of values of these forces might also be useful.
  5. Images in Figure 6 are not clear. Labelling the different particles will be helpful in understanding the images.
  6. Please explain how the separation efficiency of the device was calculated. Did you use analysis of intensity of the particles? How many experiments were conducted for each case?
  7. Does the computational model consider the same particle concentration as the experiment? Please provide comparison between your computational and experimental results and justify the results.
  8. Please provide your thoughts on the future work, usability of this device for separation of cells and as a device.
  9. Please proof read the manuscript for grammatical errors, incomplete sentences and spelling errors.

Author Response

Submission Number: micromachines- 1533176

Dear Editor:

We would like to thank you for giving us a chance to modify the paper (micromachines-1533176) entitled "Continuous particle separation driven by 3D Ag-PDMS electrodes with dielectric electrophoretic force coupled with inertia force", and thank the reviewers’ constructive suggestions for us to improve quality of the paper. We have studied all comments carefully and made a revision, which we wish to meet with approval. The changes are marked with the yellow background in the revised manuscript. The following text lists the original comments (black sentences) and the replies (red sentences). Please feel free to contact me if any additional information is needed.

Sincerely yours,

Professor Binzhen Zhang

Reviewer 2 Report

This manuscript entitled “Continuous particle separation driven by 3D Ag-PDMS electrodes with dielectric electrophoretic force coupled with inertia force” by Xiaohong Li et al. reports a continuous cell separation microfluidic device by coupling dielectrophoresis (DEP) force to inertial force. The idea is interested. There are several raised questions need to be clarified before publishing this manuscript. 

  1. The consist of the microfluidic device is unclear. The authors mentioned that “The main channel was designed with the contraction and expansion structure to provide inertial force, which was fabricated with PDMS. The microchannel sidewall electrode Ag-PDMS was designed to produce a highly uniform electric field, which could effectively improve the sorting efficiency.” In the Figure 1, it is hard to understand. The PDMS layer and Ag-PDMS layer looks overlapping.

  1. From the section of “2.3. Fabrication of the microfluidic chip” and Figure 2, it seems the whole microfluidic structure is Ag-PDMS, and bond to patterned ITO glass substrate to become the microfluidic device. It means that all the 3D electrodes are connected. Although the authors provide the sinusoid signals from independent ITO electrodes on glass substrate, the 3D Ag-PDMS electrode is entire, how does the signals apply to two side walls independently. Please the authors clarify it.
  2. The authors still not clearly show the mechanism how inertial force and DEP force to make different sizes of particles separate. It may be better to provide schematics to explain that.
  3. It is well-known that when the fluid flow through contraction-expansion structure, the secondary flow (Dean flow) will separate different sizes of particles, even without DEP force. The authors may need to explain how much contribution of DEP force in this study (DEP improvement the separation in this study). The authors may also need to provide the results that only apply pure inertial force and without DEP force for comparison.

Author Response

(The authors gave the same response as above.)

Round 2

Reviewer 1 Report

Hello,

             Authors have addressed most of the comments. Here are some changes which I would like to suggest to improve the quality of the work:

  1. The authors should include a brief description of the optimization process for choosing the electrode design in the manuscript.
  2. The authors should provide quantitative comparison of forces in the new Figure 6. This will help to understand the magnitude of each force in comparison to the other.
  3. Comments regarding the future work should be included in the manuscript.

Author Response

Submission Number: micromachines- 1533176

Dear Editor:

We would like to thank you for giving us a chance to modify the paper (micromachines-1533176) entitled "Continuous particle separation driven by 3D Ag-PDMS electrodes with dielectric electrophoretic force coupled with inertia force", and thank the reviewers’ constructive suggestions for us to improve quality of the paper. We have studied all comments carefully and made a revision, which we wish to meet with approval. Please see the attachment.

Sincerely yours,

Professor Binzhen Zhang

Reviewer 2 Report

The authors appropriately revised the manuscript and answered my question. I suggest that this manuscript can be published in current stage.

Author Response

Submission Number: micromachines- 1533176

Dear Editor:

We would like to thank you for giving us a chance to submit the paper (micromachines-1533176) entitled "Continuous particle separation driven by 3D Ag-PDMS electrodes with dielectric electrophoretic force coupled with inertia force", and thank the reviewers’ approval of our work.

Sincerely yours,

Professor Binzhen Zhang
